# Knowledge of Human Mpox (Monkeypox) and Attitude towards Mpox Vaccination among Male Sex Workers in China: A Cross-Sectional Study

**DOI:** 10.3390/vaccines11020285

**Published:** 2023-01-28

**Authors:** Yuanyi Chen, Yuwei Li, Leiwen Fu, Xinyi Zhou, Xinsheng Wu, Bingyi Wang, Xin Peng, Yinghui Sun, Qi Liu, Yi-Fan Lin, Yinguang Fan, Hongbo Jiang, Xiaojun Meng, Huachun Zou

**Affiliations:** 1School of Public Health (Shenzhen), Sun Yat-sen University, Shenzhen 518107, China; 2School of Public Health, Anhui Medical University, Hefei 230032, China; 3School of Public Health, Guangdong Pharmaceutical University, Guangzhou 510006, China; 4Wuxi Municipal Center for Disease Control and Prevention, Wuxi 214023, China

**Keywords:** human mpox (monkeypox), male sex workers (MSW), knowledge of mpox, attitude towards mpox vaccination, China

## Abstract

Background: Re-emerging human mpox (previously known as monkeypox) is spreading around the world. According to existing studies, the current mpox pandemic mainly affects men who have sex with men (MSM), including male sex workers (MSW). Our study aimed to assess mpox knowledge and attitude towards mpox vaccination among MSW in China. Methods: A web-based, cross-sectional survey was conducted in August 2022. We collected participants’ socio-demographic characteristics and knowledge with 15 knowledge items related to mpox. Modified Bloom’s cut-off points of 80% (total score > 12) was used to indicate good knowledge. Multivariable regression analysis was used to assess factors of mpox knowledge and attitude towards mpox vaccination. Results: A total of 154 MSW were recruited (age: median = 22, interquartile range, IQR = 12). Of the 154 MSW, 49.4% had good knowledge of mpox, and 63.0% were willing to be vaccinated against mpox. We found that good knowledge was associated with being single [adjusted odds ratio (AOR) = 2.46, 95% confident interval (CI) (1.22–4.87)], being unemployed [5.01, 1.21–20.70] and willingness to be vaccinated [2.51, 1.14–5.54]. Willingness to get vaccinated was related to age [1.06, 1.00–1.12], chronic diseases history [8.53, 1.01–71.68], and agreement with “priority for high-risk groups if mpox vaccine is in short supply” [2.57, 1.01–6.54]. Conclusions: We found that MSW had suboptimal mpox knowledge and a high willingness to be vaccinated against mpox. MSW who are single and willing to be vaccinated may have good knowledge of mpox. These findings underscore the necessity of providing health education on mpox among MSW. When the mpox vaccine is in short supply, priority should be given to high-risk groups, such as MSW.

## 1. Introduction

Human mpox (previously known as monkeypox) is caused by mpox virus (MPXV) and is a viral zoonosis [1]. MPXV is an enveloped double-stranded DNA virus that belongs to Orthopoxvirus genus of the Poxviridae family [1]. Symptoms of mpox are similar to those seen in smallpox patients [1]. The symptoms include skin rash (usually located on or near the genitals or anus and other areas), fever, chills, swollen lymph nodes, exhaustion, muscle aches and backache, headache, and respiratory symptoms [2]. However, compared with mpox, smallpox was more easily transmitted and more often fatal [1]. It has been hard to clinically distinguish mpox from chickenpox, a herpesvirus infection [3]. By October 2022, the virus had reached 109 locations altogether, resulting in 75,568 laboratory-confirmed cases and 34 deaths [4]. In recognition of this situation, the World Health Organization (WHO) declared mpox a Public Health Emergency of International Concern (PHEIC) on 23 July 2022 [5]. 

Men who have sex with men (MSM) were the most affected group of the current mpox outbreak [6]. Globally, MSM account for the great majority of all mpox cases, and 94% of 2891 were reported among MSM as mpox cases in the United States (up to 17 July 2022) [7]. Of the 10,912 male mpox cases with known sexual orientation, 96% self-identified as MSM from 45 countries and areas throughout the European Region (up to 25 October 2022) [8]. The transmission rate is high in MSM. Two of the three determinants (the probability of transmission, the number of contacts, and the duration of infectivity) of the basic transmission rate (R0) are high: (a) multiple sex partnership is common among MSM; (b) the probability of transmission is high. Misdiagnosis may occur in MSM. Many confirmed mpox cases in MSM population do not show typical clinical symptoms [9]. The mpox rash and some sexually transmitted diseases (STDs) share similar manifestations [9].

Male sex workers (MSW) are a subgroup of MSM who are characterized by their high number and frequency of male partnerships [10], which may render them at a very high risk for mpox transmission. The occupational health risk that MSW face relate to harm through violence from clients or pimps, factors associated with the use of drugs and mental health, and the acquisition of sexually transmitted infections (STIs) [3]. As is commonly known, once the sexually transmitted bacteria or viruses have entered the body, the infection may progress into STDs. China has a large MSW population [11]. Two vaccines (ACAM2000 and JYNNEOS) are recommended for pre-exposure prophylaxis (PrEP) of Orthopoxvirus infection among people at risk of such exposures [12]. However, these mpox vaccines are not currently available in China. 

The first imported mpox case in mainland China, reported on 16 September 2022, was a 29-year-old Chinese man who had sex with other men in Berlin during his visit to Germany and subsequently was confirmed as a case [13]. Since then, no confirmed cases of mpox have been reported among citizens of mainland China, and most of the new cases are tourists from African and non-African countries [14]. These cases, usually males, can be picked up by disease screening at international airports [14]. However, simple disease checks or simple fever screening and history taking at immigration points may not be enough to diagnose mpox [14]. Moreover, since the world announced the elimination of smallpox in the 1980s, China has stopped vaccination against smallpox for more than 30 years [15]. People born after this period lack immunity and belong to the vulnerable group [15]. Even people who have been vaccinated against vaccinia are at some risk because their immunity will decrease over time [15], and, especially, MSW in China may be at imminent risk of imported mpox transmission.

When considering the feasibility of vaccination, it is essential to understand the awareness and acceptability among target populations [16]. Our study aimed to understand the knowledge of mpox and attitude towards mpox vaccination among MSW.

## 2. Methods

The reporting of this study conforms to the Strengthening the Reporting of Observational Studies in Epidemiology (STROBE) statement and guidelines for reporting observational studies (Appendix A).

### 2.1. Participants and Procedures

We conducted a cross-sectional study design with purposive sampling among MSW on 1–31 August 2022. We used an anonymous online questionnaire for data collection. The participants were asked to fill out a questionnaire on the survey tool, Sojump (Wenjuanxing) (Changsha Ranxing Information Technology Co., Ltd., Changsha, Hunan, China), and we distributed the questionnaire via social media platforms, such as WeChat. The inclusion criteria were as follows: (1) males aged 16 years and older; (2) had ever provided paid sex services (receptive and/or insertive) to other men; (3) agreed to participate in this study. The questionnaire instrument was tested before the formal survey began, and minor amendments were made for better understanding. 

### 2.2. Data Collection and Quality Control

The specific steps of data collection were as follows: First, our research team got in touch with the frontline investigators in Wuxi, Jiangsu province, China. Second, the main researcher explained the study procedure to the investigators and then sent a link to the electronic questionnaire via WeChat (Tencent Holdings Co., Ltd., Shenzhen, Guangdong, China). Investigators know a fair amount about the characteristics of the MSW population. Thirdly, investigators explained the purpose of the study to MSW in hotspot areas, obtained informed consent, and distributed electronic questionnaires by purposive sampling. Hotspots are areas such as streets, bars, hotels, or massage parlors in Wuxi. Finally, each participant completed the survey anonymously.

In the electronic questionnaire link, a WeChat IP only allowed participants to fill in the questionnaire once so as to avoid repeating questionnaires. Meanwhile, two common-sense questions ((1). Is Shanghai, Beijing, or Nanjing the capital city in China? (2). Is yellow, white or black the most common skin color among Chinese people?)) unrelated to the purpose of the study were placed in the survey to eliminate the possibility that the questions might not be answered seriously. 

### 2.3. Study Variables

Knowledge of mpox and attitude towards the mpox vaccine were determined as response variables in this study. The first one was to investigate participants’ knowledge of mpox and was measured by 15 questions (Appendix A) with “Agree and Disagree”. The knowledge of mpox was measured by scores ranging from 0 to 15. The second one was to survey participants’ attitude towards getting the mpox vaccine and was assessed by “Willing to get mpox vaccine?” with “Yes or No” responses. 

We assessed the explanatory variables that could plausibly affect the two response variables: socio-demographic characteristics and mpox-related information. Social-demographic characteristics included age, gender identity, marital status, sexual orientation, current region, educational level, employment, monthly income, history of chronic diseases, self-reported STD history, anxiety symptoms, and depression symptoms. For anxiety symptoms, we applied the 7-item Generalized Anxiety Disorder-7 (GAD-7), and individuals with 5–21 scores were identified as having anxiety. For depression symptoms, we applied the 9-item Patient Health Questionnaire (PHQ-9), and individuals with 5–27 scores were identified as having depression symptoms. Mpox-related information are presented in Appendix A. For participants’ knowledge of mpox, its specific questions were also the explanatory variables of participants’ attitude towards getting mpox vaccine.

### 2.4. Data Analysis

Modified Bloom’s cut-off points were used to categorize the knowledge into two levels: good (scores 80%–100%, i.e., if a participant correctly answered more than 12 out of the total 15 questions) and poor (scores 0%–79%) [17]. 

Categorical variables were expressed as frequencies and constituent ratios, and continuous variables were expressed as mean and standard deviation (SD). All variables that were showed statistical significance in the univariable analysis with a *p*-value of ≤0.05 were entered into a multivariable logistic regression model. A multivariable logistic regression was used to assess correlates of respondents’ mpox knowledge and attitude towards mpox vaccination. Bar plots were used to describe the potential reasons proposed by respondents in two aspects: (a) reasons behind believing that China will or will not become an endemic country and (b) reasons behind being unwilling to get the mpox vaccine. Data analyses and graph development were conducted using R 4.2.1 (R Core Team, Vienna, Austria).

## 3. Results

### 3.1. Socio-Demographic Characteristic

A total of 170 MSW were recruited. Six participants were younger than 16 years old and ten gave incorrect answer to the common-sense questions, 154 participants were included in the analysis, resulting in a response rate of 90.6%. All participants were from Wuxi. The median age of participants was 22 (interquartile range, IQR = 12) years. One hundred and forty-eight (96.1%) were identified as cisgender male and six (3.9%) as transgender female. Eighty-three (53.9%) were not single. Ninety-one (59.1%) were heterosexual. Sixty-one (39.6%) had received college or above education. One hundred and thirty-nine (90.3%) of them were from urban places, while fourteen (9.1%) were categorized as unemployed (Table 1).

### 3.2. Whether China Would Become A Mpox Endemic Country

All participants had considered whether China would become a mpox endemic country, and 127 (82.5%) thought that China would become a mpox endemic country. Mpox could be easily spread domestically, due to the large population in China being chosen by most of the respondents. Those holding the opposing opinion (27, 17.5%) thought that China’s health system could quickly respond to the mpox outbreak (Figure 1).

### 3.3. Factors Associated with Mpox Knowledge Level

Among the 154 participants, 76 (49.4%) of them had good mpox knowledge. As shown in Table 2, only six factors had a significant effect on good mpox knowledge and, from these factors, through a multivariable analysis, only three of them were significantly associated with good mpox knowledge. The multivariable regression analysis indicated that those participants who were willing to receive the mpox vaccine were 2.51 times [adjusted odds ratio (AOR) = 2.51, 95% Confident Interval (CI) (1.14–5.54)] more likely to have good knowledge than those who were not willing to receive the mpox vaccine. Unemployed participants were 5.01 times [AOR = 5.01, 95% CI (1.21–20.70)] more likely to have good knowledge. Single participants were 2.46 times more likely to have good knowledge [AOR = 2.46, 95% CI (1.22–4.97)].

### 3.4. Attitude towards Mpox Vaccination and Correlates

Of those who responded (154), 97 (63%) were willing to get the mpox vaccine. Among the 57 participants who were unwilling to get the mpox vaccine, most of them, 37 (82%), were worried about insufficient vaccine supply. A few of them, 3 (7%), would not get vaccinated because doctors/other medical experts had not recommended vaccination (Figure 2).

The multivariable analysis indicated that for every one year added to the participants’ age, 6% were more likely to get the mpox vaccine [AOR = 1.06, 95% CI (1.00–1.12)]. Participants who had chronic diseases history were 8.53 times more likely to get the mpox vaccine [AOR = 8.53, 95% CI (1.01–71.68)]. The odds of being willing to get the mpox vaccine in participants who thought high-risk groups should be prioritized if the mpox vaccine is in short supply were 2.57 times those who thought otherwise [AOR = 2.57, 95% CI (1.01–6.54)] (Table 3).

## 4. Discussion

Our study found that nearly half of MSW in China had good knowledge of mpox, and over six out of ten participants were willing to get vaccinated. Single participants and those who were willing get the mpox vaccine were associated with good knowledge. Age and participants who agreed with “high-risk groups should be prioritized if mpox vaccine is in short supply” were associated with having the willingness to get the mpox vaccine. We also found that unemployed MSW were more likely to have good mpox knowledge, and MSW with a chronic disease history were more likely to be willing to take mpox vaccination.

Our data indicated that nearly half of MSW in China had poor knowledge of mpox. However, this may be because only one confirmed mpox case had been reported in mainland China. Even people with a high level of medical knowledge had insufficient knowledge of mpox. For example, Indonesian general practitioners (GP) had a low level of mpox knowledge, and, using the modified Bloom’s 70% cut-off point, only 36.5% had good mpox knowledge [18]. MSW and other similar populations also showed a low level of knowledge about other diseases. As for MSM, the survey on HIV/AIDS knowledge among Chinese MSM college students showed that only 42.9% had good knowledge of HIV/AIDS [19]. Similarly, 37.2% of participants in an American web-based survey of gay, bisexual, and other MSM had good knowledge of PrEP [20]. A study showed that less than a quarter of female sex workers (FSW) had heard of HPV at baseline, and all were willing to be vaccinated against HPV after being educated [21]. It is important to improve the knowledge level of mpox and health in the MSW population through knowledge intervention, such as carrying out publicity and education by pushing mpox knowledge links and playing promotional videos on websites and chat applications that are frequently used among the MSW population.

We found that single participants had better knowledge of mpox. The possible explanation could be that unmarried individuals tend to be younger. Moreover, they are more technology-savvy and good at using the Internet, which could lead to better access to the latest information of emerging infectious diseases. Another study about general practitioners reported similar findings, where younger individuals had higher odds of having good knowledge than their older counterparts, and the younger generation had better access to mpox information on the Internet [18]. 

The majority of MSW (63%) in China were willing to get the mpox vaccine. This rate is very close to that among MSM in France (60%) [22]. A systematic review, meta-analysis, and theoretical framework indicated that across all studies (78 studies), the average vaccine acceptability was 63% in MSM [23]. The vaccine acceptability rate is also very close to the rate of 52.8% of willingness to receive COVID-19 vaccine among HIV-infected MSM in mainland China (2021) [24]. In a vaccination program for MSW, 60% of MSW completed hepatitis B vaccination in England (2003) [25]. Interestingly, the willingness to vaccinate against mpox was relatively low among medical staff. It is even very low among Czech healthcare workers, with an acceptance rate of 8.8%. However, since the source of information on infectious disease outbreaks has a predictable impact on epidemic awareness and crop information, the Czech Ministry of Health is the most commonly used, and a healthcare worker poor, evidence-based medical practice may have an impact on health care workers’ willingness to vaccinate [26]. We can see that MSM, including MSW, had a relatively good attitude towards vaccine acceptance. In this case, if the MSW population need to get the mpox vaccine in the future, we may obtain a higher vaccination completion rate with less knowledge and education.

Our study found that participants willing to receive the mpox vaccine tended to have high mpox knowledge scores. As shown in a multicenter survey, parents with a history of out-of-pocket vaccination had higher knowledge of the HPV vaccine (OR = 1.43) [27]. A study also found that among Peruvian FSW, the results of a human papillomavirus vaccine trial showed that an HPV vaccination program may also lead to increased FSW knowledge [28]. However, another study found no such relationship between mpox vaccination and mpox knowledge [29]. This conflicting finding may have been influenced by factors such as: (1) the distribution of samples in our study; (2) whether the study assessed real-world or hypothetical vaccine acceptance; (3) the items used to measure knowledge and ways of classifying knowledge. People willing to be vaccinated pay more attention to health issues and are accustomed to actively gaining knowledge related to disease prevention so they have a better grasp of relevant knowledge.

Our study found that older MSW were more willing to be vaccinated against mpox. This is echoed by a study among GP that older adults are more receptive to vaccines [30]. We also found that some MSW supported the priority of high-risk populations under insufficient vaccine supply, and they were found to be more acceptable to mpox vaccination. High social interactivity of MSM contributes to an increased risk of mpox transmission in this population [9]. When the vaccine supply is limited, it is essential to ensure vaccination in key populations, such as MSW. An observational study found that MSM and FSW may be a solid target group in future HIV vaccine trails, as they were highly willing (77%) to get HIV vaccine trials [31]. Once mpox vaccine is applied, the information about its safety should be regularly made public. However, vaccination alone is not enough to end the mpox outbreak. MSM should take action to protect their sexual health, and vaccine programs should prioritize maximizing equitable access [32].

Findings from our study are subject to several limitations. Firstly, we used purposive sampling. Purposive sampling is subjective, and the estimation accuracy and sample representativeness are poor. Secondly, the sample size of our study was small. Due to the fact that the MSW population is very hidden, the number of people we could contact was small, and the number of participants was limited. A further limitation of the study was information bias. To ensure the quality of our study, we addressed quality control. We provided standard training to the investigators and adjusted the questionnaires through the pre-survey. The researchers introduced the project in detail to the respondents before filling in the questionnaire, which reduced bias and ensured the credibility of the data. In the later analysis stage, investigators screened participants’ answers to the questionnaire to ensure its quality. We acknowledge that we may have missed factors that may be related to good knowledge and willingness to vaccinate among MSW.

## 5. Conclusions

We found that MSW in China had an insufficient general knowledge of mpox and a high willingness to vaccinate against mpox. Good knowledge is associated with being single and willing to get the mpox vaccine. Vaccination willingness is related to age and the recognition that high-risk groups should be vaccinated first when vaccine supplies are insufficient. It is important to improve the capacity of MSW on mpox-related knowledge, therefore increasing this population’s health awareness. When the mpox vaccine is in short supply, priority should be given to high-risk groups, such as MSW.

## Figures and Tables

**Figure 1 vaccines-11-00285-f001:**
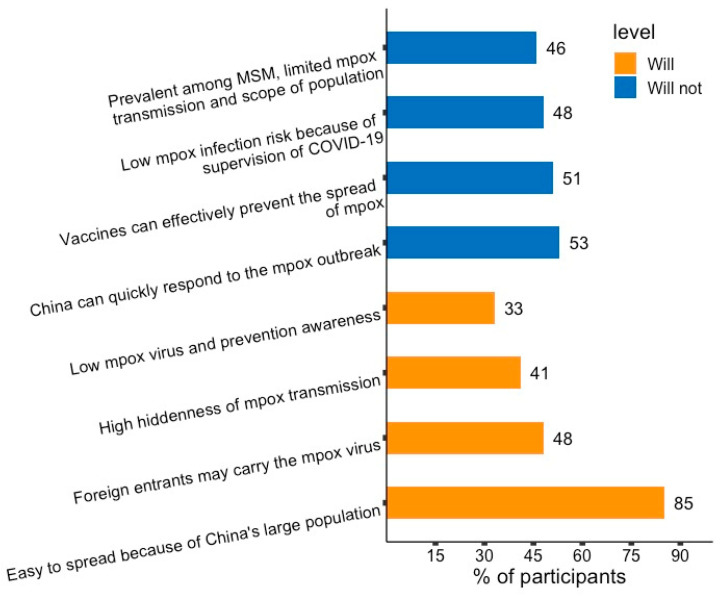
Reasons behind believing that China will or will not become an endemic country among male sex workers in China. Orange bar = reasons participants, 127 (82.5%) thought China will become an endemic country. Blue bar = reasons participants, 27 (17.5%) thought China won’t become an endemic country.

**Figure 2 vaccines-11-00285-f002:**
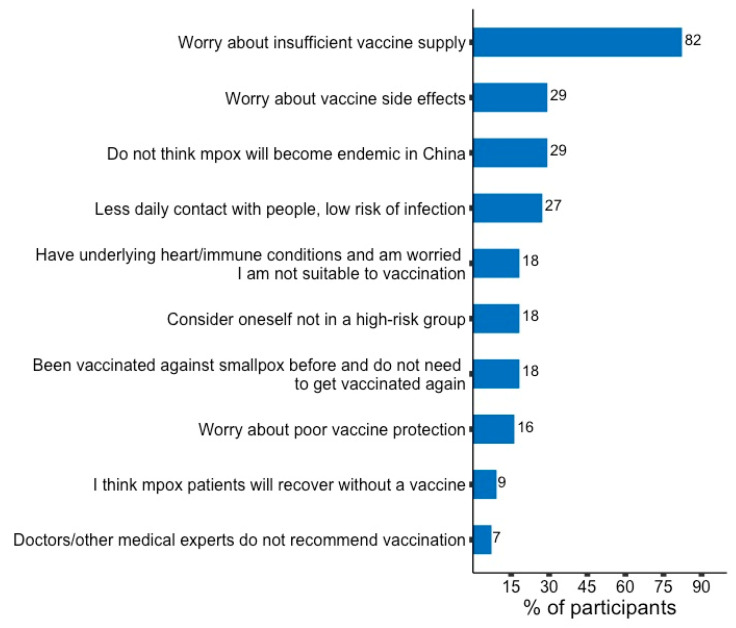
Reasons behind being unwilling to get mpox vaccine among male sex workers in China.

**Table 1 vaccines-11-00285-t001:** Socio-demographic characteristics of male sex workers in China.

Variables	Category	Frequency	Percentage
Gender identity	Cisgender male	148	96.1
	Transgender female	6	3.9
Marital status	Not single	83	53.9
	Single	71	46.1
Sexual orientation	Heterosexual	91	59.1
	Non-heterosexual	63	40.9
Educational level	College or above	61	39.6
	High school or below	93	60.4
Region	Urban	139	90.3
	Rural	15	9.7
Employment	Employed or retired	140	90.9
	Unemployed	14	9.1
Monthly income (CNY)	<5000	41	26.6
	>9990	27	17.5
	5000–9999	57	37.0
	No income	29	18.8
History of chronic diseases	No	123	79.9
	Yes	31	20.1
Self-reported STD history	No	9	5.8
	Yes	145	94.2
Have anxiety symptoms	No	139	90.3
	Yes	15	9.7
Have depression symptoms	No	125	81.2
	Yes	29	18.8

Notes: CNY = Chinese Yuan Renminbi; STD = sexually transmitted diseases.

**Table 2 vaccines-11-00285-t002:** Factors associated with mpox knowledge level among male sex workers in China.

Variables	Category	Poor Knowledge (*N* = 78)	Good Knowledge (*N* = 76)	AOR * (Multivariable)
Willing to receive mpox vaccination	No	33 (42.3%)	24 (31.6%)	
	Yes	45 (57.7%)	52 (68.4%)	**2.51 (1.14–5.54, *p* = 0.023)**
Employment	Employed or retired	75 (96.2%)	65 (85.5%)	
	Unemployed	3 (3.8%)	11 (14.5%)	**5.01 (1.21–20.70, *p* = 0.026)**
History of chronic diseases	No	59 (75.6%)	64 (84.2%)	
	Yes	19 (24.4%)	12 (15.8%)	0.43 (0.15–1.29, *p* = 0.133)
Marital status	Not single	49 (62.8%)	34 (44.7%)	
	Single	29 (37.2%)	42 (55.3%)	**2.46 (1.22–4.97, *p* = 0.012)**
Sexual orientation	Heterosexual	41 (52.6%)	50 (65.8%)	
	Non-heterosexual	37 (47.4%)	26 (34.2%)	0.51 (0.21–1.26, *p* = 0.146)
Gender identity	Cisgender male	76 (97.4%)	72 (94.7%)	
	Transgender female	2 (2.6%)	4 (5.3%)	4.77 (0.73–31.03, *p* = 0.102)

Notes: * AOR: Adjusted Odds Ratio.

**Table 3 vaccines-11-00285-t003:** Factors associated with willingness to obtain mpox vaccination among male sex workers in China.

Variables	Category	No (*N* = 57)	Yes (*N* = 97)	AOR * (Multivariable)
Age	Mean ± SD	21.8 ± 7.0	29.1 ± 10.6	**1.06 (1.00–1.12, *p* = 0.035)**
Educational level	Undergraduate or above	17 (29.8%)	44 (45.4%)	
	High school or below	40 (70.2%)	53 (54.6%)	0.94 (0.40–2.21, *p* = 0.887)
History of chronic diseases	No	56 (98.2%)	67 (69.1%)	
	Yes	1 (1.8%)	30 (30.9%)	**8.53 (1.01–71.68, *p* = 0.049)**
Ever heard of MPXV	No	30 (52.6%)	24 (24.7%)	
	Yes	27 (47.4%)	73 (75.3%)	1.61 (0.22–12.03, *p* = 0.642)
Paid attention to mpox outbreak information	Never	37 (64.9%)	40 (41.2%)	
	Always	9 (15.8%)	17 (17.5%)	0.65 (0.16–2.71, *p* = 0.558)
	Often	11 (19.3%)	40 (41.2%)	1.30 (0.35–4.80, *p* = 0.692)
Accessed mpox information from the Internet	No	33 (57.9%)	35 (36.1%)	
	Yes	24 (42.1%)	62 (63.9%)	1.25 (0.22–7.18, *p* = 0.802)
Accessed mpox information from traditional media	No	53 (93%)	69 (71.1%)	
	Yes	4 (7%)	28 (28.9%)	2.93 (0.75–11.45, *p* = 0.122)
Belief that China will become a mpox endemic country	No	52 (91.2%)	75 (77.3%)	
	Yes	5 (8.8%)	22 (22.7%)	2.40 (0.71–8.17, *p* = 0.161)
Belief that high-risk groups should be prioritized if mpox vaccine is in short supply	Disagree	21 (36.8%)	13 (13.4%)	
	Agree	36 (63.2%)	84 (86.6%)	**2.57 (1.01–6.54, *p* = 0.048)**

Notes: * AOR: Adjusted Odds Ratio. The adjusted variables are: age; educational level; history of chronic diseases; ever heard of MPXV; paid attention to mpox outbreak information; accessed mpox information from the Internet; accessed mpox information from traditional media; belief that China will become a mpox endemic country; belief that high-risk groups should be prioritized if mpox vaccine is in short supply.

## Data Availability

The data that support the findings can only be obtained by the questionnaire designer on online platform Sojump (Wenjuanxing) (Changsha Ranxing Information Technology Co., Ltd., Changsha, Hunan, China, https://www.wjx.cn. accessed on 1 August 2022), and are not publicly available in consideration of ethics.

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
