# Peer review of "Knowledge of Human Mpox (Monkeypox) and Attitude towards Mpox Vaccination among Male Sex Workers in China: A Cross-Sectional Study"

_vaccines, 2023, doi:10.3390/vaccines11020285_

Round 1
Reviewer 1 Report
The manuscript is well written and focused on an interesting issue, considering a high-risk population.
I have some minor concern.
First, the Authors reports only the first case of monkeypox which was diagnosed in China at the time of the investigation but they should also highlight which is the current epidemiological situation since some month passed in the meantime.
Furthermore, in the methods they should better explain how MSW were specifically selected considering that the online questionnaire could have been potentially filled in by everyone.
Finally, since the regression analysis is performed to find possible predictors, knowledge should be considered as a possible predictor of willingness to be vaccinated and included in that analysis, not the contrary.
Author Response
Authors’ response: Thank you for your comments, and here are the answers to the questions:
First, in order to explained the current epidemiological situation, we added some sentences as follows:
“Since then, no confirmed cases of monkeypox have been reported among citizens of inland China, and most of the new cases are tourists from African and non-African countries.4 These cases, usually males, can be picked up by disease screening at international airports.4 But simple disease checks or simple fever screening and history taking at immigration checkpoints may not be enough to diagnose the disease.4 Moreover, since the world announced the elimination of smallpox in the 1980s, China has stopped vaccination against smallpox for more than 30 years.5 People born after this period lack immunity and belong to the vulnerable group.5 Even people who have been vaccinated against vaccinia are at some risk because their immunity will decrease over time.5” (Line 88-97, Page 2)
Second, we have "inclusion criteria" in the "2.1. Participants and procedures" section of the article. And the "Data collection" section has been added to further explain the recruitment strategy. Moreover, our research is based on purposive sampling, which means the selection and determination of research objects are according to the research objectives and subjective judgment of the situation. Since our investigators have a good understanding of the characteristics of the MSW population, we can only investigate eligible MSW who we can reach.
For the last comment, we thought we should actually answer the following two questions:
1- Why knowledge is not used as a predictive factor of vaccination willingness?
2- Why the willingness to vaccinate is as a predictive factor of knowledge?
First, we would like to answer the second one, like we discussed in line 339-346, willingness to vaccinate correlates with knowledge levels. Because this study is a cross-sectional study, regression analysis can only determine whether there is an association, but cannot infer causality, that’s why we interpreted this result with caution.
For the first one, by incorporating all 15 items of knowledge into the equation as variables, and fitting the equation, then the optimal equation is obtained. The variables are shown in Table 3, however, there are no items of knowledge. The correlation between knowledge level as a variable and willingness to vaccinate was not explored. Because when knowledge level was used as a dependent variable, knowledge level was obtained by summing 15 knowledge items. Knowledge level and 15 items of knowledge could not occur in the same equation as variables. When exploring the relationship between vaccination willingness and other factors, the 15 small items as variables can make full use of information and explore more meaningful variables, although the results are not satisfactory.
Reviewer 2 Report
1. Please use the official name (mpox) in the entire manuscript.
2. Remove "Wuxi" from the title.
3. Line 60: Where is the reference for this important claim "China has a large MSW population"?
4. The keywords should be better suggested according to the MeSH/NCBI: https://www.ncbi.nlm.nih.gov/mesh/
5. In the Introduction section, you may consider reflecting on the differential diagnosis of mpox - especially when compared to smallpox and chickenpox.
6. The Introduction section needs to reflect on the occupational risks that MSW are exposed to, especially STDs.
7. The study should be reported according to the STROBE guidelines for cross-sectional studies. Please cite the STROBE guidelines in your Methods section.
Suggested ref (optional):
https://doi.org/10.1016/S0140-6736(07)61602-X
8. Please upload the STROBE checklist as a supplementary file. https://www.strobe-statement.org/checklists/
9. The recruitment strategy was unclear. How did you ensure that the participants were MSW or even MSM?
10. What are the psychometric properties of the used questionnaire? Please report the validity and reliability values.
11. Why were 16 participants excluded?
12. Table 3, which variables did you adjust for?
13. The Discussion section may benefit from reflecting on additional similar studies.
Suggested ref (optional):
https://pubmed.ncbi.nlm.nih.gov/36560432/
Author Response
- Please use the official name (mpox) in the entire manuscript.
Authors’ response: The issues have been corrected.
- Remove "Wuxi" from the title.
Authors’ response: The issues have been corrected.
- Line 60: Where is the reference for this important claim "China has a large MSW population"?
Authors’ response: We have added some references in the revised manuscript.
- The keywords should be better suggested according to the MeSH/NCBI: https://www.ncbi.nlm.nih.gov/mesh/
Authors’ response: We appreciate the reviewer for providing the suggestion. The issues have been corrected in the revised manuscript.
- In the Introduction section, you may consider reflecting on the differential diagnosis of mpox - especially when compared to smallpox and chickenpox.
Authors’ response: Thank you for your suggestion. We added a sentence as follows:
“However, compared with monkeypox, smallpox was more easily transmitted and more often fatal.1 It has been hard to clinically distinguish monkeypox from chickenpox, a herpesvirus infection.2” (Line 41-43, Page 1)
- The Introduction section needs to reflect on the occupational risks that MSW are exposed to, especially STDs.
Authors’ response: Thank you for your suggestion. We added sentences as follows:
“The occupational health risks that MSW may face relate to harm through violence from clients or pimps, factors associated with the use of drugs and mental health, and the acquisition of sexually transmitted infections (STIs).2, 3 Moreover, once the sexually transmitted bacteria or viruses have entered the body, the infection may progress into STDs that may cause further harm to MSW.” (Line 78-82, Page 2)
- The study should be reported according to the STROBE guidelines for cross-sectional studies. Please cite the STROBE guidelines in your Methods section.
Suggested ref (optional):
https://doi.org/10.1016/S0140-6736(07)61602-X
Authors’ response: Thank you for your suggestion.
According to the STROBE guidelines, we added a missed section in the revised manuscript, the content is as follows:
“2.2. Data collection and quality control
The specific steps of data collection are as follows: First, our research team got in touch with the local investigators in Wuxi, Jiangsu province, China. Second, the principle investigator explained the study procedure to the local investigators and then send a link to the electronic questionnaire via WeChat. Local investigators know a fair amount about the characteristics of the MSW population. Thirdly, local investigators explained the purpose of the study to MSW in hotspot areas, obtained informed consent, and distributed electronic questionnaires by purposive sampling. Hotspots are areas such as streets, bars, hotels, or massage parlors in Wuxi. Finally, each participant completed the survey anonymously.
In the electronic questionnaire link, a WeChat IP can only fill in the questionnaire once, so as to avoid repeated questionnaires. Meanwhile, two common-sense questions (1. Is Shanghai, Beijing, or Nanjing the capital city in China? 2. Is yellow, white or black the most common skin color among Chinese people?) unrelated to the purpose of the study were placed in the survey to eliminate the possibility that the questions might not be answered seriously.” (Line 142-156, Page 3)
We added the STROBE guidelines as follows:
“The reporting of this study conforms to the Strengthening the Reporting of Observational Studies in Epidemiology (STROBE) statement, guidelines for reporting observational studies (Supplementary File 1).” (Line 103-105, Page 2)
- Please upload the STROBE checklist as a supplementary file. https://www.strobe-statement.org/checklists/
Authors’ response: We added the STROBE checklist as a supplementary file.
- The recruitment strategy was unclear. How did you ensure that the participants were MSW or even MSM?
Authors’ response: We have "inclusion criteria" in the "2.1. Participants and procedures" section of the article. And the "Data collection" section has been added to further explain the recruitment strategy. Moreover, our research is based on purposive sampling, which means the selection and determination of research objects are according to the research objectives and subjective judgment of the situation. Since our investigators have a good understanding of the characteristics of the MSW population, we can only investigate eligible MSW who we can reach.
- What are the psychometric properties of the used questionnaire? Please report the validity and reliability values.
Authors’ response: Thank you for your question. Here are the validity and reliability values of GAD-7 and PHQ-9:
GAD-7(Cronbach's alpha =0.967; KMO=0.903, Bartlett’s test of sphericity, P<0.05)
PHD-9(Cronbach's alpha =0.964; KMO=0.915, Bartlett’s test of sphericity, P<0.05)
In conclusion, the GAD-7 and PHD-9 are scales with good psychometrics and can serve as tools for anxiety screening and depression screening among MSW populational.
These two scales (GAD-7, PHD-9) were only used as an intermediary to judge whether the research subjects were anxious or depressed, and each item in the scale was not used as a variable, so it seems that the reliability and validity may be not suitable for reporting in the article.
- Why were 16 participants excluded?
Authors’ response: Thank you for your comment. We have added the following sentences in the revised manuscript:
“6 participants were younger than 16 years old, and 10 answered the common-sense questions incorrectly, therefore, 154 included in this survey giving a response rate of 90.6%.” (Line 207-208, Page 4)
- Table 3, which variables did you adjust for?
Authors’ response: Thank you for your question.
AOR is obtained after incorporating multiple variables into multivariable regression, and the AOR value of a variable is the OR value after correction of all other variables in the table.
The adjusted variables are: age, educational level, history of chronic diseases, ever heard of mpox virus, paid attention to mpox outbreak information, accessed mpox information from the Internet, accessed mpox information from traditional media, belief that China will become a mpox endemic country, belief that high-risk groups should be prioritized if mpox vaccine is in short supply.
- The Discussion section may benefit from reflecting on additional similar studies.
Suggested ref (optional):
https://pubmed.ncbi.nlm.nih.gov/36560432/
Authors’ response: Thank you for your suggestion. We have enriched the discussion section in revised manuscript.